# Theoretical and Experimental Investigations of Large Stokes Shift Fluorophores Based on a Quinoline Scaffold

**DOI:** 10.3390/molecules25112488

**Published:** 2020-05-27

**Authors:** Barbara Czaplińska, Katarzyna Malarz, Anna Mrozek-Wilczkiewicz, Aneta Slodek, Mateusz Korzec, Robert Musiol

**Affiliations:** 1Institute of Chemistry, Faculty of Science and Technology, University of Silesia in Katowice, 75 Pułku Piechoty 1A, 41-500 Chorzów, Poland; czaplinska.basia@gmail.com; 2A. Chełkowski Institute of Physics, Faculty of Science and Technology, University of Silesia in Katowice, 75 Pułku Piechoty 1, 41-500 Chorzów, Poland; katarzyna.malarz@us.edu.pl (K.M.); anna.mrozek-wilczkiewicz@us.edu.pl (A.M.-W.); 3Silesian Center for Education and Interdisciplinary Research, University of Silesia in Katowice, 75 Pułku Piechoty 1A, 41-500 Chorzów, Poland; 4Institute of Chemistry, Faculty of Science and Technology, University of Silesia in Katowice, Szkolna 9, 40-007 Katowice, Poland; aneta.slodek@us.edu.pl (A.S.); mateusz.korzec@us.edu.pl (M.K.)

**Keywords:** green fluorophores, quinoline, large Stokes shift, intramolecular charge transfer, cell imaging, DFT calculations

## Abstract

A series of novel styrylquinolines with the benzylidene imine moiety were synthesized and spectroscopically characterized for their applicability in cellular staining. The spectroscopic study revealed absorption in the ultraviolet–visible region (360–380 nm) and emission that covered the blue-green range of the light (above 500 nm). The fluorescence quantum yields were also determined, which amounted to 0.079 in the best-case scenario. The structural features that are behind these values are also discussed. An analysis of the spectroscopic properties and the theoretical calculations indicated the charge-transfer character of an emission, which was additionally evaluated using the Lippert–Mataga equation. Changes in geometry in the ground and excited states, which had a significant influence on the emission process, are also discussed. Additionally, the capability of the newly synthesized compounds for cellular staining was also investigated. These small molecules could effectively penetrate through the cellular membrane. Analyses of the images that were obtained with several of the tested styrylquinolines indicated their accumulation in organelles such as the mitochondria and the endoplasmic reticulum.

## 1. Introduction

Quinoline is an attractive scaffold for designing new fluorescent agents due to its small molecular size; the presence of nitrogen, which creates good coordination properties; its ability to form hydrogen bonds, and a fair synthetic availability [1]. At the same time, quinoline itself is also known as a fluorophore unit, which further increases its potential. It is also widely used as a core molecular fragment in a broad range of drugs and biologically active substances [2,3,4]. For this reason, it has been claimed to be a privileged structure [5]. Our research group is interested in synthesizing and investigating quinoline derivatives [6,7,8]. In recent years, we obtained a series of new styrylquinolines with anticancer and antifungal properties [9,10]. Besides active compounds, some of the newly synthesized structures also turned out to be good fluorophores [11], which is in agreement with many reports concerning the application of quinoline’s structure in biological imaging. Due to their planar and rigid aromatic structure, quinolines interact with proteins and nucleic acids. Therefore, they were used for DNA and RNA imaging with good results (Figure 1A) [12,13,14,15]. Both activities (therapeutic and diagnostic) of a styrylquinoline derivative were also exploited by Staderini et al. to stain the Aβ amyloid plaques in Alzheimer’s disease (Figure 1B) [16]. Highly fluorescent quinolines can also detect both gram-positive and gram-negative bacteria (Figure 1C) [17]. Quinoline derivatives, especially 8-hydroxy substituted are particularly good chelators for metal ions, and therefore, this type of probes have dominated this field of chemosensing [18]. Among them are the Zn^2+^ sensors, which have very good properties (Figure 1D–I) [19,20,21,22]. Indeed, the first and most commonly known fluorescent zinc ion indicator was 8-hydroxyquinoline, which was used in 1968, for detection in human plasma [23].

Another appealing feature of the quinoline-related compounds is their tendency to have an intramolecular charge transfer characteristic (ICT) [24,25]. A plausible reason for this is the electron-accepting nature of the moiety because of a well-defined dipole moment. Compounds with ICT exhibit different properties in the ground and excited states, which is caused by an electronic redistribution after photoexcitation. This transient alternation leads to large changes in dipole moments and to the relaxation of the structure. The energy difference between the Frank–Condon state and the ICT state is the reason for the large Stokes shift, which is preferred in most fluorescence methods [26].

In biological applications, particularly fluorescent microscopy, the large stokes shift which is usually over 80 nm are particularly useful for the reduction of chromatic aberrance and increase of resolution [27]. Stimulated emission depletion (STED) microscopy is a technique that utilizes res-shifted light pulse for reducing the excitation of the molecules peripheral to the excitation spot [28]. Consequently, their excitation is reduced, which allows us to overcome the diffraction barrier in high resolution images. An excellent review on large stokes shift dyes in such STED microscopy was published recently [29].

Another application of large stokes shift dyes is the Fӧrster resonance energy transfer (FRET) microscopy, a technique based on pair dyes that can undergo direct excitation of an acceptor by the adjacent donor molecules [30]. An effective dye pair needs overlapping in the emission and excitation spectra of donor and acceptor molecule, respectively, which inevitably leads to cross-excitation of the acceptor upon the donor, resulting in false-positive results. Large stokes shift dyes are helpful to overcome this problem, increasing the threshold of detection. Moreover, compounds of this type have attracted much attention because of their possible applications in photoelectronic devices, as non-linear materials, emitters, or in organic light-emitting diodes (OLED) [31,32,33]. Since the ICT state emission is usually strongly dependent on environmental parameters like the solvent polarity or viscosity, push–pull compounds are also widely used as chemical sensors [34,35]. Based on our experience with the styrylquinoline compounds, we decided to substitute this scaffold with a Schiff base moiety (Figure 2).

Schiff bases are frequently reported for their antifungal, antibacterial, antiproliferative, anti-inflammatory, antimalarial, and antiviral activity [36,37,38]. They are also used as metal ions or pH sensors [39,40]. On the other hand, Schiff bases are relatively rarely exploited for their fluorescent potential. However, the unique properties of the imine bond, namely its polarized electron density and propensity to form a tautomeric structure, make it useful for designing compounds with ICT characteristics. Thus, in the present report, we provide the chemical, physicochemical, and optical profiles of the newly obtained compounds, together with the theoretical data and biological assays.

## 2. Results and Discussion

### 2.1. Spectroscopic Properties

#### 2.1.1. Absorption Properties

The absorption and fluorescence of all newly synthesized compounds were investigated using UV–VIS spectroscopy. When selecting the solvents for the experiments, various properties were taken into account, such as polarity and the ability to create hydrogen bonds, since these might influence fluorescence. The results that were obtained in this study are presented in Table 1. The obtained compounds exhibited a good excitation/emission profile. All absorption spectra consisted of two bands and the maxima of the more red-shifted ones were over 360 nm, which is desirable in cell imaging. Wavelengths shorter than 350 nm carry enough energy to destroy the complex protein structure, which interferes with the observation process and, therefore, are unsuitable for long-term experiments [41]. Absorption wavelengths above 360 nm enable avoidance of any autofluorescence from naturally occurring molecules whose excitation profile embrace higher-energy wavelengths, such as 270 nm (tyrosine) [42], 280 nm (tryptophan) [43], or 340 nm (NAD(P)H) [44]. When the absorption spectra (Figure 3) were analyzed, a relationship between a structure and the band shift were observed. The biggest red shift was recorded for compound **3a**, which had a strong electron-donating group in its structure (N(CH_3_)_2_). Therefore, attaching an electron-donating N(CH_3_)_2_ substituent in **3a** led to a markedly enhanced extinction coefficient of the low energy band, similar to the compound **3c** bearing the OH group (Table 1). Then, the spectra maxima shifted toward the longer wavelengths, together with an increase of the donating power of the substituent that was attached to a phenyl moiety (**3a** > **3c** > **3b** > **3f** > **3e**). Apparently, compound **3d** did not fit into a relationship that could be explained on a structural basis. The phenyl vinyl substituent constituted a larger conjugated aromatic system that diluted the partitive effects and changed the overall dipole of a molecule.

#### 2.1.2. Fluorescence Properties

The fluorescence spectra were recorded in various solvents as well, which enabled the charge transfer character of the emission to be revealed. All fluorescence spectra are presented in Figure 4 and the corresponding data are listed in Table 2. As one can see, the compounds exhibited the best fluorescent properties in ethanol. The emission peaks occurred around 514 nm in ethanol for all **3a**–**f** compounds, and were intense and strongly red-shifted compared to those in the chloroform and acetonitrile and were comparable to those in DMSO (except for **3a**). On the other hand, in DMSO and acetonitrile, the intensity of the emission drastically decreased in most cases, while in the chloroform, an average intensity and blue-shifted spectra could be observed.

These data suggest that the emission spectra are strongly dependent on the solvent polarity, which probably arises from a big difference in the dipole moment between the ground and excited states [45]. At the same time, it is easily noticeable that the changes had a specific tendency, which is called positive solvatochromism—an increase of the solvent polarity induces the red shift of an emission band. The intramolecular charge transfer phenomenon is another reason for the very large Stokes shift that was observed in the synthesized compounds. The changes of the dipole moment that occurred in the excited state caused the relaxation of the solvent around an excited molecule, to obtain a state of minimum energy [46].

The smallest Stokes shift of 131 nm (6505 cm^−1^) was observed for **3a** and the largest one was 152 nm (8169 cm^−1^) for **3e**, for the measurements that were performed in ethanol. The Stokes shift is critical to the sensitivity of the fluorescence method because it enables a good detection of photons against a low background as well as the separation from an excitation source. Unfortunately, such large Stokes shifts are usually associated with small fluorescence quantum yields. These were also measured for all compounds, as presented in Table 1. The smallest quantum yield was observed for **3d** and amounted to 0.034. Whereas, for **3f**, the value was 0.079, which was the highest fluorescence efficiency that was observed. Nevertheless, all of these features together made these compounds suitable for in vitro biological imaging.

#### 2.1.3. Influence of Water on the Spectroscopic Properties

Additionally, we decided to conduct an in-depth investigation of the spectroscopic properties of the dyes in a water environment, as is typical in analytical conditions that include the cell interior. For this purpose, because of the low water solubility of our compounds, we carried out a series of measurements in ethanol, with an increasing amount of water. The sets of the relationships that were obtained through these experiments are presented in Table 3.

According to the results, water had almost no influence on the absorption spectra. The absorbance maxima of each compound did not shift, compared to their maxima in ethanol. An exception was compound **3a** whose absorption maximum was significantly blue-shifted in the 40%/60% mixture, which could be an effect of the higher protonation of the dimethylamine group in the highly protic solvent [47]. Fluorescence, on the other hand, is strongly dependent on the environmental factors, because an increased amount of water shifts the emission to the longer wavelengths. At the same time, the Stokes shifts also had higher values in the aqueous solutions, e.g., 172 nm vs. 148 nm in the case of compound **3f**, which was in accordance with the increase of fluorescence intensity. In light of these results, we decided to further study this phenomenon. Moreover, a recently published article that investigated the hydrolysis and aggregation processes of the Schiff bases was brought to our attention, which raised more doubts about the changes that occur in water. Due to the possibility that the compounds would degrade, we used the HPLC method for the samples that contained 25% water and 75% ethanol, and measured the composition of the mixtures after 5, 15, 30, and 60 min. The results of this study are presented in Figure 5.

For some compounds, the decomposition was quite rapid. For compounds **3b** and **3f**, only 30–40% of the product remained after one hour. On the other hand, compounds **3d** and **3e** appeared to be rather resistant to hydrolysis. Interestingly, these observations correlated with the structural features that possibly affect the dipole moment of a molecule. The methoxy- (**3b**) and ethoxy-substituted (**3f**) compounds decomposed quickly, while the more lipophilic, less polarized ones were stable. The styryl- (**3d**) or 2-fluorophenyl- (**3e**) groups did not have an increased electron density on the imine bond, which had a positive effect on stability. It should be noted here that this type of decomposition does not always disturb the applications in water-based environments, e.g., biological systems. Recently, we published a report on the application of a pH-sensitive Schiff base, as a cellular probe for staining lysosomes [48].

#### 2.1.4. Estimating the Dipole Moment (the Lippert–Mataga Equation)

In order to explain the large Stokes shifts in relation to changes in the structure in the excited states, we applied the Lippert–Mataga Equation (1) [49,50]. Estimating the difference between the excited- and ground-state dipole moments in this equation (Δµ = µ_e_ − µ_g_) exploited the relationship between a Stokes shift and the solvent polarity. This method is widely used in photophysical studies of the push–pull molecules to elucidate the internal charge transfer (ICT) characteristics [24,25,51,52]. The equation used was as follows:(1)$$\Delta v=\Delta E_{exc-em}=\frac{2{\left(\mu_{e}-\mu_{g}\right)}^{2}}{hca^{3}}\Delta f+Const.$$
(2)$$\Delta f=\frac{\varepsilon -1}{2\varepsilon +1}-\frac{n^{2}-1}{2n^{2}+1}$$
where Δ*ν* is the Stokes shift in a given solvent (cm^−1^), *h* is the Planck’s constant, *c* is the speed of light in a vacuum, and *a* is the Onsager cavity radius. The Onsager cavity radii and the ground-state dipole moments (*μ_g_*) were obtained from the quantum chemical calculation (Gaussian09, DFT/B3LYP/6-311+G(d)) [41,53]. The Δ*f* is the orientation polarizability of the solvent, which measures both the electron mobility and dipole moment of a solvent molecule. Δ*f* was calculated using Equation (2), where *ε* is the dielectric constant of the solvent and *n* is the optical refractive index of the solvent. Figure 6 presents the Lippert–Mataga plots, which provide the slopes to the further calculations of equation:(3)$$\left(Slope=\frac{2}{hca^{3}}{\left(\Delta \mu_{eg}\right)}^{2}\right)$$
whose values are presented in Table 4, together with the results that were obtained from the equation. The calculated values of Δ*μ_eg_* were 16.7 D, 18.4 D, 16.3 D, 17.8 D, 16.6 D, and 15.8 D, respectively, for compounds **3a**–**f**. Such big differences between the dipole moments indicate the internal charge transfer (ICT) as being a process that is involved in acquiring the excited state.

### 2.2. Theoretical Calculations

#### Electronic Structure and Transition Energies

To gain a more precise insight into the spectroscopic properties of the newly synthesized compounds, theoretical calculations were performed. The Time-Dependent Density Functional Theory with the CAM-B3LYP functional and the 6-311+G(d, p) basis set (TD-DFT//CAM-B3LYP//6-311+G (d, p)) appeared to be the best method for this task. Additionally, the PCM model was used to evaluate the solvent effects. The results are presented in Table 5, which includes the characteristics (oscillator strength-f, transition energy–λ/nm together with the main orbital configurations) of the main electronic transitions that were responsible for the band maxima. Furthermore, Figure 7 presents a visualization of the border orbitals HOMO and LUMO, which were involved in the main electronic transitions. The applied method gave satisfactory results compared to the experimental data. The high values of the oscillator strengths resulted from the method that was used for the calculation. According to the theoretical data, the experimental absorption maximum of the first band was mainly assigned to the S_0_→S_1_ transition. This transition employed the HOMOs and LUMOs, whose shapes indicated a π→π* type of transition. In the case of compound **3a**, the electron density of the highest occupied orbital was located on the phenyl ring with the N(CH_3_)_3_ group, and the lowest unoccupied orbital embraced the quinoline structure. There was a distinct charge transfer between the donor and the acceptor units. A similar property could be observed in the case of the rest of the compounds. However, the charge transfer process was no longer distinguishable. The absorption spectra of all compounds consisted of two bands. The second bands were assigned to the higher transitions, such as S_0_→S_5_ or S_0_→S_6_, and used the HOMO orbitals with a lower energy (like HOMO−1, HOMO−3, and HOMO−6) and the LUMO orbitals with a higher energy (such as LUMO+1 and LUMO+2, and LUMO+3). Quantum calculations were also performed for the spin-allowed singlet→singlet (S_n_→S_0_) transitions, in order to determine the theoretical values of the fluorescence emission energy. The results are presented in Table 6. The obtained values of the emission maxima correctly reflected the experimental data.

Due to the structural relaxation, there was a decrease in the dihedral angle between the styrylquinoline plane and the phenyl ring (C38–N28–C27–C24, red color in Figure 8). In other words, the molecule became more planar. As a result of this change, the energy difference between the HOMO and LUMO also decreased and the energy of the emission was lower than the energy of the excitation, which resulted in a larger Stokes shift (Figure 9). These relationships were in agreement with the Jablonski diagram. The changes in the dihedral angles are presented in Table 7 and are depicted in Figure 8.

### 2.3. Biological Activity and Cellular Imaging

The combination of large Stokes shifts, absorption over 350 nm, and a good fluorescent profile predestine the investigated compounds to be molecular probes for biological staining. For such applications, various physicochemical properties such as lipophilicity, protein binding, ion chelation, or halochromism could be exploited, in order to achieve specific staining effects. However, inactivity toward the biological systems is of special importance. We performed cytotoxicity tests on the human colon cancer cell line (HCT 116) and the normal human fibroblast cell line (NHDF), to evaluate the potential applications in this appealing field. The results are shown in Table 8. In general, all tested compounds had no prolonged cytotoxicity against the HCT 116 and NHDF cells, in concentrations that are useful for staining.

Incubation with synthesized compounds **3a**–**f** for 72 h did not affect the viability of the cells. This fact, together with their good spectroscopic properties, make them potentially useful as fluorescent dyes for biological specimens. However, with regards to the possibility of the decomposition of compounds in an aqueous environment, only two derivatives (**3e**,**d**) were selected for the cell imaging experiments. Additionally, for the co-localization experiments, the incubation time was reduced to 1 h, in order to minimize the risk of decomposition. The site of the accumulation of the selected compounds was examined in the HCT 116 cells, using distinct specific-organelle markers for mitochondria (MitoTracker Orange), lysosomes (LysoTracker^®^ Red DND-99), and the endoplasmic reticulum (ER-Tracker™ Red BODIPY^®^ TR Glibenclamide). Apparently, these compounds tend to accumulate in the membranous structures, and their associated organelles accumulate mainly in the mitochondria and endoplasmic reticulum (ER) (Figure 10). Moreover, there is a subtle but evident relationship between the structure of a dye and the apparent site of accumulation. According to the scientific literature, it has not yet been determined why the fluorescent probes for the ER are selective. However, compounds with an affinity to the mitochondria and ER, including compounds **3d** and **3e** have a few common features, which might be crucial for their behavior in cells. Namely, these dyes usually have a medium-size aromatic system (20–30 atoms) and a moderate (or higher) lipophilicity (**3d**—logP = 7.1, **3e**—logP = 5.3). Furthermore, imines might be partially protonated in a cell, which might increase their affinity to the negatively charged mitochondrial membrane. The cationic character of a compound was also a typical feature for the ER dyes [54].

## 3. Experimental Method

### 3.1. Materials and Basic Measurements

All reagents were purchased from Sigma-Aldrich. The TLC experiments were performed on aluminum-backed silica gel 40 F254 plates (Merck, Darmstadt, Germany). The plates were illuminated under UV (254 nm and 365 nm). The melting points were determined on an Optimelt MPA-100 apparatus (SRS, Stanford, CA, USA). The structures of the newly obtained compounds were confirmed by NMR spectroscopy. All NMR spectra were recorded in deuterated DMSO-d_6_ as the solvent on a Bruker AM-series (Bruker BioSpin Corp., Karlsruhe, Germany). The working frequency for each compound was known. Chemical shifts were reported in ppm (δ), against the internal standard Si(CH_3_)_4_. The mass spectra were performed on a WATERS LCT Premier XE system (high-resolution mass spectrometer with a Time Of Flight (TOF) analyzer).

### 3.2. Synthesis and Characterization

#### 3.2.1. Synthesis of the Starting Material

*Step 1: 2-[(E)-2-(4-nitrophenyl)ethenyl]quinoline (**1**):* The synthesis was performed according to the procedure described in [10]. The quinaldine (10 mmol) and 4-nitrobenzaldehyde (10 mmol) were mixed in acetic anhydride and heated for 20 h at 130 °C. The excess solvent was evaporated in vacuo and the obtained solid was purified using column chromatography, with dichloromethane as the eluent. The product was obtained as a bright yellow solid with a yield of 73%, and a melting point of 167 °C. ^1^H NMR (400 MHz, DMSO) δ 8.41 (d, *J* = 8.6 Hz, 1H), 8.27 (d, *J* = 8.6 Hz, 2H), 8.06–7.99 (m, 4H), 7.97 (d, *J* = 6.4 Hz, 1H), 7.94 (d, *J* = 8.6 Hz, 1H) 7.78 (t, *J* = 7.8 Hz, 1H), 7.73 (d, *J* = 16.4 Hz, 1H), 7.60 (t, *J* = 7.4 Hz, 1H). ^13^C NMR (126 MHz, DMSO) δ 155.27, 148.11, 147.36, 143.49, 137.24, 133.63, 132.22, 130.52, 129.30, 128.67, 128.35, 127.77, 127.17, 124.53, 120.82.

*Step 2: 4-[(E)-2-(quinolin-2-yl)ethenyl]aniline (**2**):* In a round-bottom flask, compound **1** was placed in ethanol, together with anhydrous SnCl_2_, at a molar ratio of 1:5. The mixture was heated for 2 h at a temperature of 70 °C, under nitrogen. Then, the mixture was cooled, placed in a flask with ice, and neutralized with a 5% solution of sodium bicarbonate (NaHCO_3_). Next, the extraction with ethyl acetate was performed, the organic layer was washed with brine and dried over anhydrous sodium sulfate (Na_2_SO_4_). The product was obtained as a red solid with a yield of 61% and a melting point of 174 °C. ^1^H NMR (400 MHz, DMSO) δ 8.26 (d, *J* = 8.6 Hz, 1H), 7.93 (d, *J* = 8.4 Hz, 1H), 7.89 (d, *J* = 8.0 Hz, 1H) 7.77 (d, *J* = 8.6 Hz, 1H), 7.73–7.69 (m, 1H), 7.67 (d, *J* = 16.1, 1H), 7.50 (t, *J* = 7.4 Hz, 1H), 7.42 (d, *J* = 8.3 Hz, 2H), 7.13 (d, *J* = 16.2 Hz, 1H), 6.61 (d, *J* = 8.3 Hz, 2H), 5.54 (s, 2H). ^13^C NMR (126 MHz, DMSO) δ 157.03, 150.39, 148.22, 136.54, 135.54, 130.07, 129.21, 128.81, 128.18, 127.10, 125.95, 124.22, 123.28, 120.06, 114.33. ^13^C NMR (126 MHz, DMSO) δ 157.03, 150.39, 148.22, 136.54, 135.54, 130.07, 129.21, 128.81, 128.18, 127.10, 125.95, 124.22, 123.28, 120.06, 114.33.

#### 3.2.2. Representative Procedure for Compounds **3a**–**f**

Compound **2** and the corresponding aldehyde were placed in a round-bottom flask, at a ratio of 1:1. Then, 5 mL of ethanol and 3 drops of acetic acid were added. The mixture was exposed to microwave irradiation for 20 min in conditions of 50 W and 83 °C. The obtained compounds were filtered and washed with ethanol. The general route for the synthesis is presented in Figure 2.

*(N-{[4-(Dimethylamino)phenyl]methylidene}-4-[(E)-2-(quinolin-2-yl)ethenyl]aniline (**3a**):* The product was obtained as a yellow solid with an yield of 62% and a melting point of 202 °C. ^1^H NMR (500 MHz, DMSO) δ 8.49 (s, 1H), 8.35 (d, *J* = 8.6 Hz, 1H), 7.99 (d, *J* = 8.5 Hz, 1H), 7.95 (d, *J* = 7.3 Hz, 1H), 7.87 (d, *J* = 9.00 Hz, 1H), 7.86 (d, *J* = 16.5 Hz, 1H), 7.79–7.73 (m, 5H), 7.56 (t, *J* = 7.4 Hz 1H), 7.46 (d, *J* = 16.3 Hz, 1H), 7.27 (d, *J* = 8.4 Hz, 2H), 6.80 (d, *J* = 9.0 Hz, 2H), 3.03 (s, 6H). ^13^C NMR (126 MHz, DMSO) δ 160.31, 156.29, 153.07, 153.10, 148.24, 136.84, 134.33, 133.63, 130.88, 130.21, 129.11, 128.69, 128.22, 128.14, 127.47, 126.49, 124.40, 121.94, 120.39, 112.01. ESI–MS calcd. for C_26_H_24_N_3_ 378.1970 [M+H]^+^ found 378.1973.

*N-[(3,4-Dimethoxyphenyl)methylidene]-4-[(E)-2-(quinolin-2-yl)ethenyl]aniline (**3b**):* The product was obtained as a yellow solid with an yield of 70% and a melting point of 172 °C. ^1^H NMR (500 MHz, DMSO) δ 8.59 (s, 1H), 8.36 (d, *J* = 8.6 Hz, 1H), 8.00 (d, *J* = 8.4 Hz, 1H), 7.95 (d, *J* = 7.5 Hz, 1H), 7.87 (d, *J* = 9.0 Hz, 1H), 7.86 (d, *J* = 16.5 Hz, 1H), 7.80 (d, *J* = 8.4 Hz, 2H), 7.76 (t, *J* = 7.7 Hz, 1H), 7.58–7.55 (m, 2H), 7.49 (d, *J* = 16.5 Hz, 1H), 7.48 (d, *J* = 8.5 Hz, 1H), 7.32 (d, *J* = 8.4 Hz, 2H), 7.12 (d, *J* = 8.4 Hz, 1H), 3.85 (s, 6H). ^13^C NMR (126 MHz, DMSO) δ 160.66, 156.20, 152.43, 152.37, 149.51, 148.16, 136.9, 134.22, 134.14, 130.30, 129.50, 129.10, 128.77, 128.46, 128.28, 127.46, 126.59, 124.74, 122.09, 120.45, 111.79, 109.88. ESI–MS calcd. for C_26_H_23_N_2_O_2_ 395.1760 [M+H]^+^ found 395.1763.

*4-(N-{4-[(E)-2-(quinolin-2-yl)ethenyl]phenyl}carboximidoyl)phenol (**3c**):* The product was obtained as a yellow solid with an yield of 67% and a melting point of 248 °C. ^1^H NMR (500 MHz, DMSO) δ 10.17 (s, 1H), 8.54 (s, 1H), 8.36 (d, *J* = 8.6 Hz, 1H), 8.00 (d, *J* = 8.4 Hz, 1H), 7.95 (d, *J* = 7.9 Hz, 1H), 7.88 (d, *J* = 8.5 Hz, 1H), 7.86 (d, *J* = 16.5 Hz, 1H), 7.82–7.73 (m, 5H), 7.56 (t, *J* = 7.5 Hz, 1H), 7.47 (d, *J* = 16.3 Hz, 1H), 7.29 (d, *J* = 8.3 Hz, 2H), 6.90 (d, *J* = 8.5 Hz, 2H). ^13^C NMR (126 MHz, DMSO) δ 161.26, 160.54, 156.22, 152.62, 148.16, 136.92, 134.20, 134.02, 131.30, 130.30, 129.09, 128.74, 128.33, 128.27, 127.97, 127.45, 126.57, 122.03, 120.43, 116.17. ESI–MS calcd. for C_24_H_19_N_2_O 351.1497 [M+H]^+^ found 351.1500.

*N-[(2E)-3-phenylprop-2-en-1-ylidene]-4-[(E)-2-(quinolin-2-yl)ethenyl]aniline (**3d**):* The product was obtained as a yellow solid with an yield of 68% and a melting point of 195 °C. ^1^H NMR (400 MHz, DMSO) δ 8.49 (d, *J* = 8.9 Hz, 1H), 8.36 (d, *J* = 8.6 Hz, 1H), 8.00 (d, *J* = 8.4 Hz, 1H), 7.95 (d, *J* = 8.0 Hz, 1H), 7.89 (d, *J* = 8.8, 1H), 7.87 (d, J = 16.8 Hz, 1H), 7.81–7.70 (m, 5H), 7.51 (s, 1H), 7.49, (d, *J* = 16.4 Hz, 1H), 7.45–7.40 (m, 4H), 7.30 (d, *J* = 8.3 Hz, 2H), 7.20 (dd, *J* = 16.0, 8.9 Hz, 1H).^13^C NMR (126 MHz, CD_2_Cl_2_) δ 161.47, 155.89, 152.07, 148.34, 144.13, 136.21, 135.69, 134.55, 133.53, 129.63, 129.60, 129.08, 128.90, 128.66, 128.43, 128.16, 127.53, 127.36, 126.04, 121.49, 119.60. ESI–MS calcd. for C_26_H_21_N_2_ 361.1705 [M+H]^+^ found 361.1707.

*N-[(2-fluorophenyl)methylidene]-4-[(E)-2-(quinolin-2-yl)ethenyl]aniline (**3e**):* The product was obtained as a yellow solid with an yield of 68% and a melting point of 124 °C. ^1^H NMR (400 MHz, DMSO) δ 8.85 (s, 1H), 8.37 (d, *J* = 8.6 Hz, 1H), 8.14 (t, *J* = 7.1 Hz, 1H), 8.00 (d, *J* = 8.4 Hz, 1H), 7.96 (d, *J* = 8.0 Hz, 1H), 7.89 (d, *J* = 8.8, 1H), 7.87 (d, *J* = 16.8 Hz, 1H), 7.83 (d, *J* = 8.3 Hz, 2H), 7.76 (t, *J* = 7.6 Hz, 1H), 7.67–7.54 (m, 2H), 7.51 (d, *J* = 16.3 Hz, 1H), 7.43–7.33 (m, 4H). ^13^C NMR (126 MHz, DMSO) δ 163.65, 161.64, 156.11, 153.80, 151.80, 148.14, 136.96, 135.09, 134.22 (d, *J* = 8.4 Hz), 133.99, 130.32, 129.01 (d, *J* = 23.5 Hz), 128.80, 128.31 (d, *J* = 8.5 Hz), 127.49, 126.64, 125.42, 123.88 (d, *J* = 8.8 Hz), 122.26, 120.46, 116.69 (d, *J* = 20.6 Hz). ESI–MS calcd. for C_24_H_18_FN_2_ 353.1448 [M+H]^+^ found 353.1452.

*N-[(2-ethoxyphenyl)methylidene]-4-[(E)-2-(quinolin-2-yl)ethenyl]aniline (**3f**):* The product was obtained as a yellow solid with an yield of 55% and a melting point of 167 °C. ^1^H NMR (500 MHz, DMSO) δ 8.91 (s, 1H), 8.36 (d, *J* = 8.6 Hz, 1H), 8.04 (dd, *J* = 7.7, 1.8 Hz, 1H), 8.00 (d, *J* = 8.5 Hz, 1H), 7.96 (d, *J* = 8.1 Hz, 1H), 7.89 (d, *J* = 8.5 Hz, 1H), 7.87 (d, *J* = 16.5 Hz, 1H), 7.80 (d, *J* = 8.4 Hz, 2H), 7.78–7.74 (m, 1H), 7.57 (dt, *J* = 8.0, 6.9, 1.1 Hz, 1H), 7.54–7.51 (m, 1H), 7.49 (d, *J* = 16.4 Hz, 1H), 7.29 (d, *J* = 8.4 Hz, 2H), 7.17 (d, *J* = 8.2 Hz, 1H), 7.07 (t, *J* = 7.5 Hz, 1H), 4.18 (q, *J* = 7.0 Hz, 2H), 1.40 (t, *J* = 7.0 Hz, 3H). ^13^C NMR (126 MHz, DMSO) δ 159.23, 156.18, 156.14, 152.85, 148.20, 136.91, 134.51, 134.12, 133.75, 130.26, 129.12, 128.80, 128.65, 128.25, 127.49, 127.43, 126.58, 124.55, 121.96, 121.12, 120.44, 113.54, 64.52, 14.76. ESI–MS calcd. for C_26_H_23_N_2_O 379.1810 [M+H]^+^ found 379.1806.

### 3.3. Biological Properties

#### 3.3.1. Cell Culture

The human colon carcinoma cell line HCT 116 was obtained from ATCC. The cells were grown as monolayer cultures in 75 cm^2^ flasks (Nunc) in Dulbecco’s modified Eagle’s medium with antibiotic gentamicin (200 μL/100 mL medium). DMEM was supplemented with 12% heat-inactivated fetal bovine serum (Sigma). The cells were cultured under standard conditions, at 37 °C in a humidified atmosphere at 5% CO_2_.

#### 3.3.2. Cytotoxicity Studies

The cells were seeded in 96-wells plates (Nunc) at a density of 5 × 10^3^ cells/well and incubated at 37 °C for 24 h. The assay was performed, following a 72 h incubation with varying concentrations of the compounds (**3a**–**f**) that were being tested. Then, 20 µL of the CellTiter 96^®^AQueous One Solution-MTS (Promega) was added to each well (with 100 µL DMEM without phenol red) and incubated for 1 h at 37 °C. The optical densities of the samples were analyzed at 490 nm, using a multi-plate reader Synergy 4 (BioTek, BioTek Instruments, Winooski, VT, USA). The results were calculated as the inhibitory concentration (IC_50_) values using the GraphPad Prism 7, and are expressed as a percentage of the control. Each individual compound was tested in triplicates in a single experiment, with each experiment being repeated three times.

#### 3.3.3. Cellular Staining

The cells were seeded onto glass slides at a density of 3 × 10^5^ cells/slide and were incubated at 37 °C for 24 h. Then, the medium was removed and the solutions of **3a**–**f**, at a concentration of 25 μM, were added. The cells were then further incubated for 1 h and 2 h, under standard conditions—at 37 °C in a humidified atmosphere at 5% CO_2_. After incubation, the cells were washed three times with PBS and fixed with 3.7% paraformaldehyde, for 10 min. The cellular staining with compounds being tested was observed using a Nikon Eclipse Ni-U microscope, equipped with a Nikon Digital DS-Fi1-U3 camera and the corresponding software (Nikon, Tokyo, Japan).

#### 3.3.4. Subcellular Localization

The cells were seeded onto glass slides at a density of 3 × 10^5^ cells/slide and incubated at 37 °C for 24 h. Then, the medium was removed and the solutions of **3a**–**f**, at a concentration of 25 μM, were added. The cells were then further incubated for 1 h and 2 h. After incubation, the cells were rinsed with PBS (pH 7.2) and the staining procedures were performed according to the protocols of the providers. In brief, a serum-free medium that contained MitoTracker^®^ Orange (100 nM, 30 min incubation, Molecular Probes), ER-Tracker™ Red BODIPY^®^ TR Glibenclamide (1 µM, 30 min incubation, Molecular Probes), and LysoTracker^®^ Red DND-99 (500 nM, 1 h incubation, Molecular Probes) or Hoechst 33342 (6.5 µM, 15 min incubation, Molecular Probes) were added. After staining with organelle-specific trackers, the cells were washed three times with PBS and fixed with 3.7% paraformaldehyde for 10 min. The subcellular localization was observed using a Nikon Eclipse Ni-U microscope equipped with a Nikon Digital DS-Fi1-U3 camera and the corresponding software. The analysis and processing of the images were performed using an Image J 1.41 (Wayne Rasband, National Institutes of Health, Bethesda, MD, USA).

### 3.4. Spectroscopic Studies

The absorption and fluorescence spectra were measured at room temperature in a 10 mm quartz cell, using a U-2900 spectrophotometer (Hitachi) and an F-7000 spectrofluorometer (Hitachi), respectively. Due to the low solubility of the compounds in the applied solvents, the solutions that were used in the test of the solvatochromic behavior contained 1% of DMSO, to improve the solubility. The quantum yields of fluorescence were determined using the absolute method at room temperature, using an integrating sphere with a solvent as a blank on an FLS-980 spectrophotometer. The compounds were excited at the wavelength that corresponded with the absorption wavelength of the compounds.

### 3.5. Theoretical Calculations

The geometrical parameters of an investigated compound and its electronic properties in the ground and excited states were calculated within the DFT and TD–DFT approximations, using the Gaussian 09 software package [53] with the CAM-B3LYP exchange-correlation functional [55,56,57] and the 6-311+G (d, p) basis set. The effects of the solvent were evaluated using the polarizable continuum model (PCM) [58,59] in which a cavity was created via a series of overlapping spheres [60] with the standard dielectric constants (ε) of 24.55 for ethanol. The lowest 50 singlet–singlet vertical electronic excitations based on the B3LYP optimized geometries and the optimization of the first excited states were computed using the time-dependent density functional theory (TD–DFT) formalism [61]. The visualization of the molecular orbitals and analysis of the energy levels were performed using the Chemissian software.

## 4. Conclusions

A series of styrylquinoline dyes with a Schiff base fragment was designed and synthesized, and their physicochemical parameters and spectroscopic features were determined. All compounds had a high Stokes shift and strong solvatochromism. Along with the calculations that were carried out accordingly, these suggested an intramolecular charge-transfer mechanism of fluorescence. All obtained compounds appeared to be non-toxic in a broad range of concentrations, during the in vitro experiments. Styrylquinolines maintained their fluorescent properties in biological systems, thus allowing for staining applications to use them. Their selective accumulation in the cellular organelles was also investigated, which indicated a preference for structures such as the mitochondria and the endoplasmic reticulum.

## Figures and Tables

**Figure 1 molecules-25-02488-f001:**
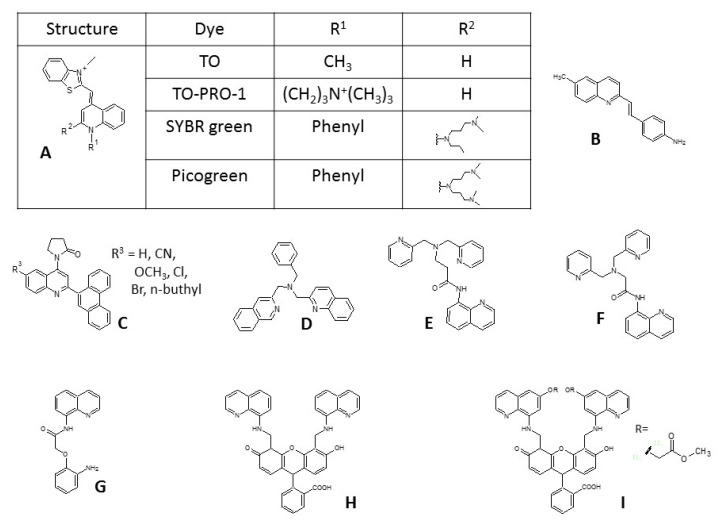
Examples of quinoline-based fluorophores and sensors. (**A**) carbocyanine dyes for DNA and RNA; (**B**) styrylquinoline for amyloid plaques; (**C**) quinoline dyes for bacterial application; (**D**) bisquinoline dye for metal detection; (**E**) and (**F**) dipicolylamine derivatives of quinoline as chemosensing dyes; (**G**) 8-aminoquinoline dye for physiological fluorescent labeling of Zn^2+^; (**H**) and (**I**) fluorescein based probes for Zn^2+^ detection.

**Figure 2 molecules-25-02488-f002:**
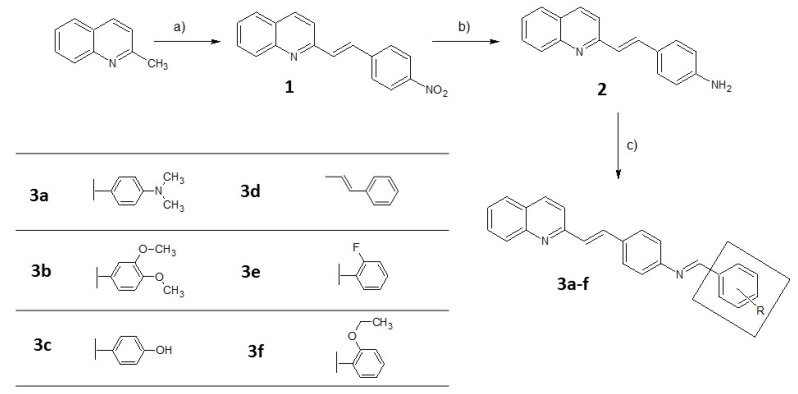
The synthetic pathway of compounds **3a**–**f**. Reagents and conditions: (**a**) Aromatic aldehyde/Ac_2_O, (**b**) SnCl_2_/EtOH, and (**c**) aromatic aldehyde/EtOH.

**Figure 3 molecules-25-02488-f003:**
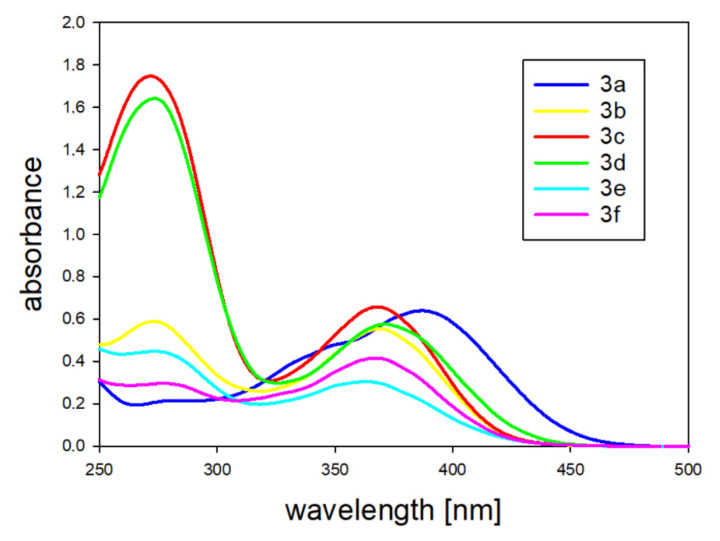
UV–Vis absorption spectra of the Schiff bases **3a**–**f** in ethanol (1.25 × 10^−5^ M).

**Figure 4 molecules-25-02488-f004:**
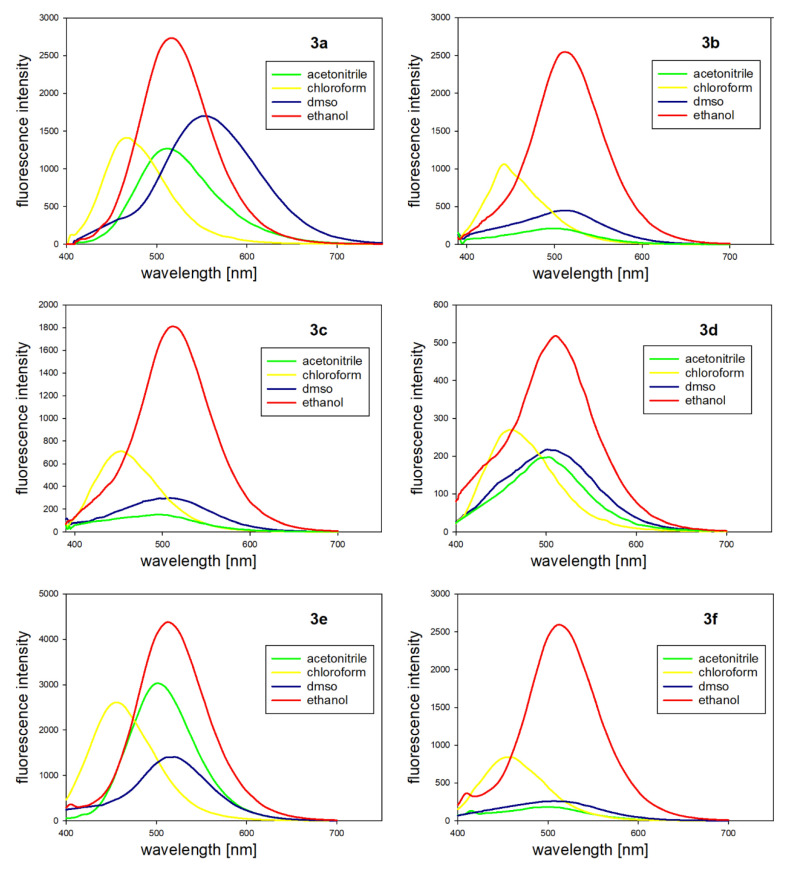
Fluorescence spectra of compounds **3a**–**f** in the various solvents (1.25 × 10^−5^ M).

**Figure 5 molecules-25-02488-f005:**
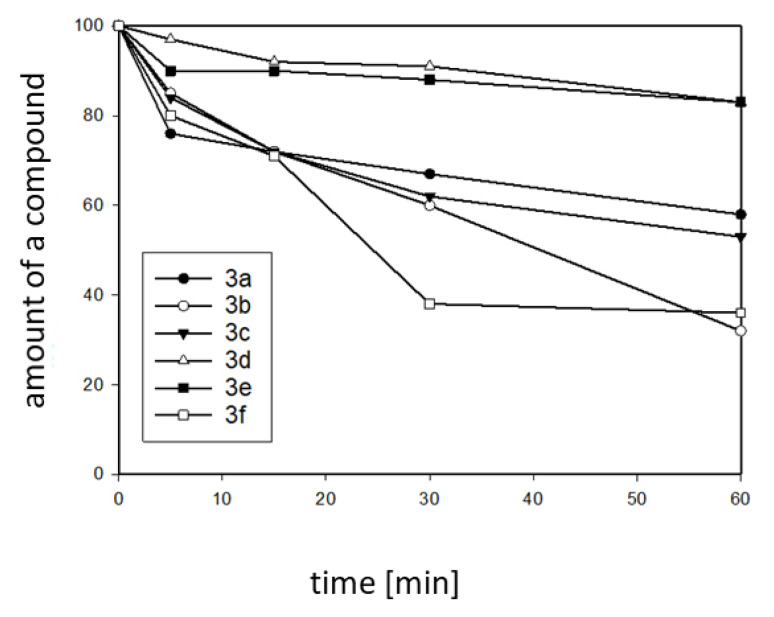
Plot presenting the decomposition rate of compounds **3a**–**f** in a water environment.

**Figure 6 molecules-25-02488-f006:**
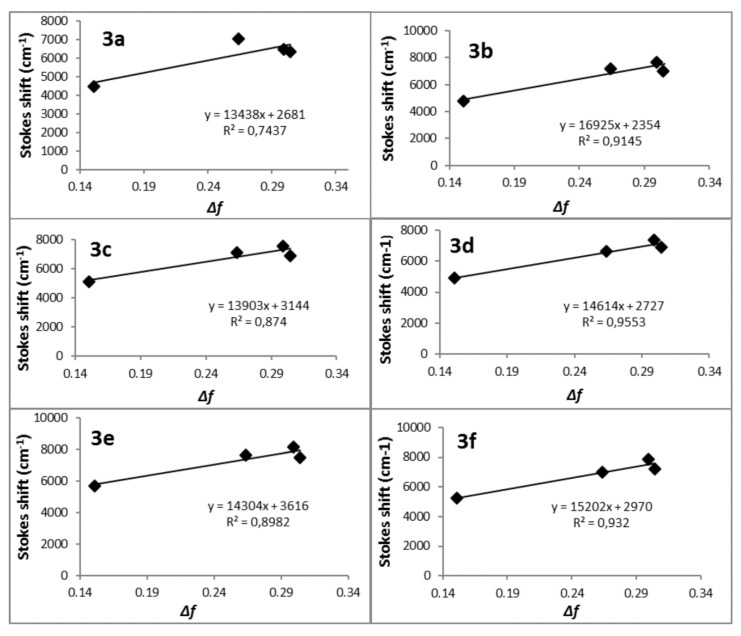
Plots of the Stokes shifts as a function of the solvent polarity parameter Δ*f* (ε, n^2^) for compounds **3a**–**f**.

**Figure 7 molecules-25-02488-f007:**
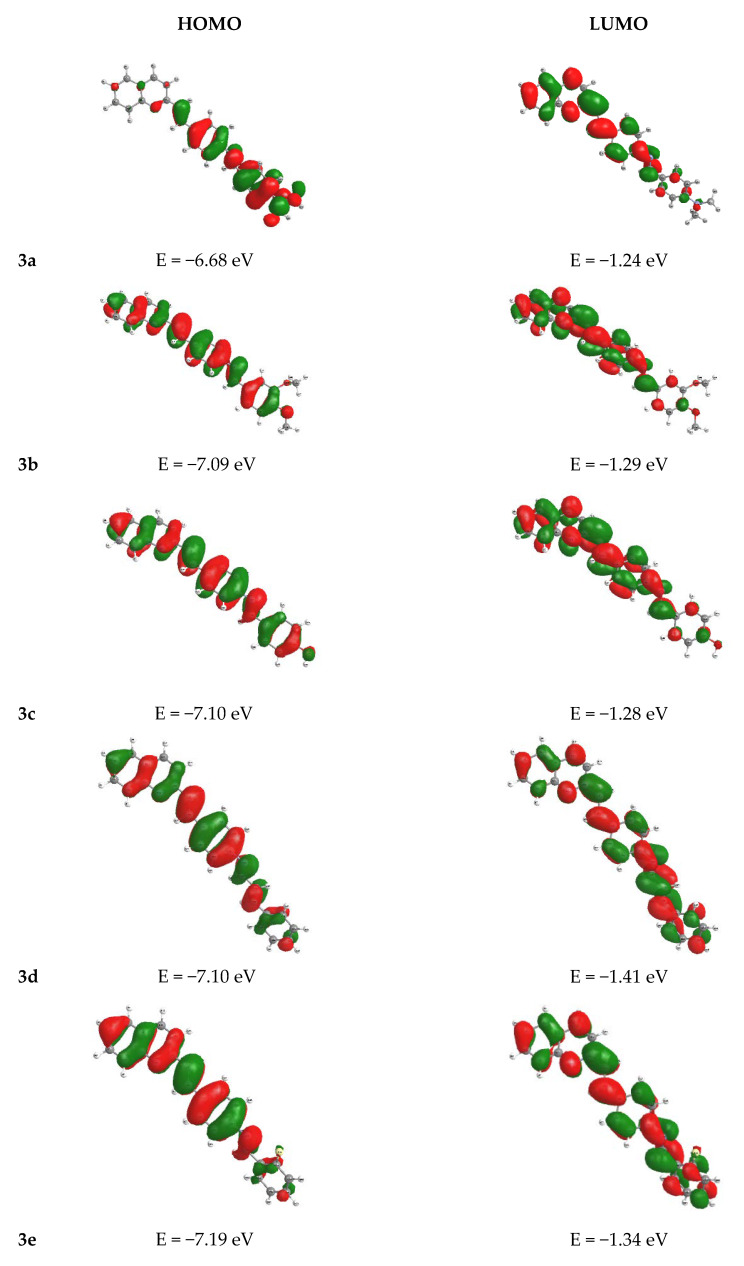
Visualization of the molecular orbitals that were involved in the main electronic transitions.

**Figure 8 molecules-25-02488-f008:**
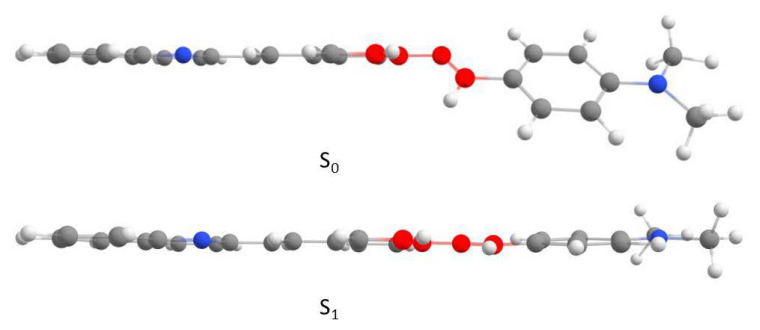
The geometry of compound **3a** in the ground and excited states, which was obtained by TD–DFT//B3LYP//6-311+G (d).

**Figure 9 molecules-25-02488-f009:**
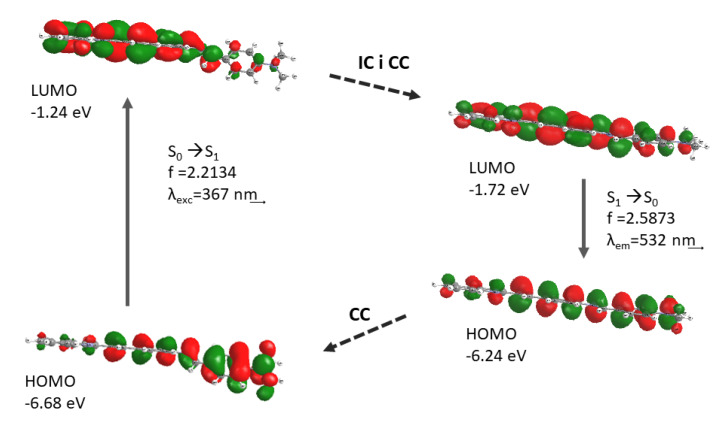
Geometry relaxation of compound **3a** after photoexcitation. The more planar structure was the reason for the decrease in the energy gap between the HOMO and LUMO orbitals. CC—configuration change and IC—internal conversion.

**Figure 10 molecules-25-02488-f010:**
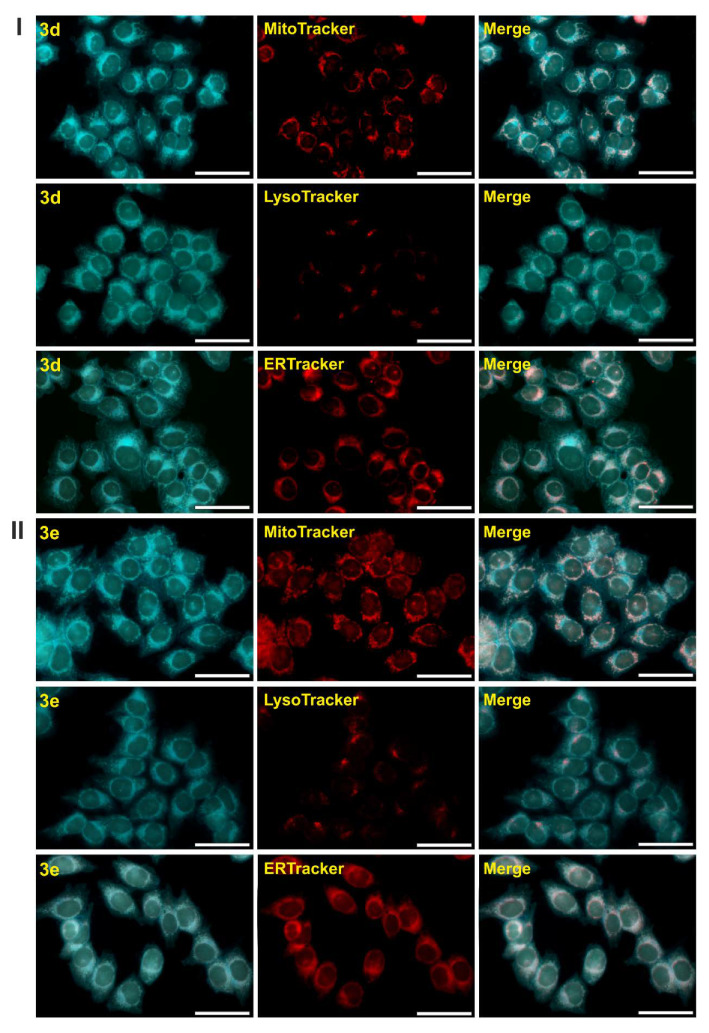
Fluorescence images of the HCT 116 cells that were incubated with **3d** (**I**) and **3e** (**II**) (both at a concentration of 25 µM), for 1 h at 37 °C. The images show a co-localization with specific-organelle trackers—the mitochondria, lysosomes, and endoplasmic reticulum. The first panel represents the fluorescence of the **3d** and **3e** compounds alone, the middle panel—the specific dyes, and the last—the merged. Scale bars indicate 50 µm. The absorption profile of the tested compounds enabled their excitation with a DAPI wavelength and the large Stokes shift helped to minimize the self-quenching effect and also boosted the signal-to-noise ratio, which in turn enabled us to obtain images of a good resolution.

**Table 1 molecules-25-02488-t001:** Fluorescent and absorption properties of the Schiff bases **3a**–**f** in ethanol.

Compound	Absorption	Emission		Stokes Shift nm/cm^−1^
	λ_max_/nm (ε/10^3^ M^−1^ cm^−1^)	λ_max_/nm	φ	
**3a**	388 (51.1) 336 (33.3)	519	0.041	131/6505
**3b**	368 (44.4) 276 (46.8)	512	0.054	144/7642
**3c**	370 (52.3) 272 (139.6)	514	0.052	144/7571
**3d**	372 (46.1) 274 (131.3)	512	0.034	140/7350
**3e**	362 (24.3) 270 (35.6)	514	0.058	152/8169
**3f**	366 (33.3) 272 (23.4)	514	0.079	148/7867

**Table 2 molecules-25-02488-t002:** Solvatochromic properties of the compounds **3a**–**f**; S–Stokes shift.

Compound	Chloroform			Acetonitrile			DMSO			Ethanol		
	λabs	λem	S	λabs	λem	S	λabs	Λem	S	Λabs	Λem	S
	nm	nm	nm/cm^−1^	nm	nm	nm/cm^−1^	nm	nm	nm/cm^−1^	nm	nm	nm/cm^−1^
**3a**	386	467	81/4493	387	513	126/6346	400	557	157/7046	388	519	131/6505
**3b**	370	450	80/4804	370	500	130/7027	374	511	137/7168	368	512	144/7642
**3c**	368	454	86/5147	367	502	135/7327	374	510	136/7130	370	514	144/7571
**3d**	374	458	84/4904	372	500	128/6881	382	512	130/6646	372	512	140/7350
**3e**	362	456	94/5694	364	501	137/7512	370	516	146/7647	362	514	152/8169
**3f**	370	459	89/5240	368	500	132/7174	374	507	133/7014	366	514	148/7867

**Table 3 molecules-25-02488-t003:** Spectroscopic properties of **3a**–**f** depending on the amount of water in a solution. S—BioTek Stokes shift.

Compound	Ethanol			Ethanol + Water			Ethanol + Water			Ethanol + Water		
				(80%/20%)			(50%/50%)			(40%/60%)		
	λabs	λem	S	Λabs	Λem	S	λabs	Λem	S	λabs	λem	S
	nm	nm	nm/cm^−1^	nm	nm	nm/cm^−1^	nm	nm	nm/cm^−1^	nm	nm	nm/cm^−1^
**3a**	388	519	131/6505	388	527	139/6797	388	534	146/7046	354	540	186/9730
**3b**	368	512	144/7642	368	524	156/8090	368	530	162/8305	366	534	168/8595
**3c**	366	514	146/7867	366	524	158/8238	366	530	164/8454	366	534	168/8595
**3d**	372	512	140/7350	372	524	152/7797	372	532	160/8084	366	536	170/8665
**3e**	362	514	152/8169	360	525	165/8730	362	529	167/8720	360	533	173/9016
**3f**	366	514	148/7867	366	522	156/8165	366	530	164/8454	362	534	172/8897

**Table 4 molecules-25-02488-t004:** The Onsager cavity radii, slopes, and changes of the dipole moments of compounds **3a**–**f**.

Compound	Onsager Cavity Radius (Å)	Slope (cm^−1^)	Δ*μ_eg_* (Debye)	*μ_g_* (Debye)	*μ_e_* (Debye)
**3a**	5.94	13,438 ± 5578	16.7	5.6	22.3
**3b**	6.00	16,925 ± 3660	18.4	3.8	22.2
**3c**	5.48	13,903 ± 3722	16.3	3.1	19.4
**3d**	5.66	14,614 ± 2235	17.8	2.3	20.1
**3e**	5.63	14,304 ± 3404	16.6	3.2	19.3
**3f**	6.27	15,202 ± 2888	15.8	3.5	19.3

**Table 5 molecules-25-02488-t005:** Electronic transition data obtained by TD–DFT//B3LYP//6-311+G (d), using a polarizable continuum model (PCM) (solvent–ethanol) for compounds **3a**–**f**, at the DFT-optimized geometry.

Compound	Electronic Transition	λ (nm)	f	Molecular Orbital	Percentage Contribution (%)	Experimental λ (nm)
**3a**	S_0_→S_1_	367	2.2134	HOMO→LUMO	57	388
	S_0_→S_6_	274	0.0800	HOMO−1→LUMO+4	55	282
**3b**	S_0_→S_1_	356	1.8851	HOMO→LUMO	86	368
	S_0_→S_5_	271	0.2781	HOMO−1→LUMO HOMO−1→LUMO+1	23 18	276
**3c**	S_0_→S_1_	355	1.8087	HOMO→LUMO	88	370
	S_0_→S_5_	267	0.3937	HOMO−1→LUMO HOMO−3→LUMO+1	23 18	272
**3d**	S_0_→S_1_	364	2.1305	HOMO→LUMO	85	372
	S_0_→S_5_	280	0.2265	HOMO−2→LUMO HOMO−6→LUMO	15 14	274
**3e**	S_0_→S_2_	354	1.6569	HOMO→LUMO	88	362
	S_0_→S_5_	265	0.4493	HOMO→LUMO+2 HOMO−1→LUMO	21 12	270
**3f**	S_0_→S_1_	358	1.7470	HOMO→LUMO	87	366
	S_0_→S_6_	265	0.2758	HOMO→LUMO+2 HOMO→LUMO+3	21 11	272

**Table 6 molecules-25-02488-t006:** Emission data obtained for **3a**–**f** by TD–DFT//B3LYP//6-311+G (d), using a PCM model (ethanol).

Compound	Electronic Transition	Λ (nm)	f	Molecular Orbital	Percentage Contribution (%)	Experimental λ (nm)
**3a**	S_0_→S_1_	532	2.5873	HOMO→LUMO	91	519
**3b**	S_0_→S_1_	502	2.2812	HOMO→LUMO	93	512
**3c**	S_0_→S_1_	501	2.2154	HOMO→LUMO	93	514
**3d**	S_0_→S_1_	540	2.4128	HOMO→LUMO	91	512
**3e**	S_0_→S_1_	-	-	-	-	514
**3f**	S_0_→S_1_	522	1.9072	HOMO→LUMO	91	514

**Table 7 molecules-25-02488-t007:** Comparison of a compound’s geometry in the ground and excited states.

	Dihedral Angle (C38-N28-C27-C24)/[°]		
Compound	Ground State S_0_	Excited State S_1_	Difference
**3a**	42.5	4.5	38.0
**3b**	42.5	15.9	26.6
**3c**	42.9	17.0	25.9
**3d**	41.7	13.3	28.4
**3e**	-	-	-
**3f**	42.7	16.0	26.7

**Table 8 molecules-25-02488-t008:** Antiproliferative activity on the HCT 116 cell line.

Compound	Cytotoxicity–IC_50_ [µM]	
	HCT 116	NHDF
**3a**	>25	>25
**3b**	>25	>25
**3c**	>25	>25
**3d**	>25	>25
**3e**	>25	>25
**3f**	>25	>25

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
