# Peer review of "Theoretical and Experimental Investigations of Large Stokes Shift Fluorophores Based on a Quinoline Scaffold"

_molecules, 2020, doi:10.3390/molecules25112488_

Round 1
Reviewer 1 Report
The authors synthesized styrylquinoline derivatives and studied their spectroscopic character and application for the cellular staining. This manuscript was well written in a wide range of fields from synthetic part to biological application. Based on these points, the reviewer considered that this manuscript is suitable for publication in Molecules after minor revision.
- Page 4 to 5, 2,2,1 synthesis of starting material; The authors should recheck the 1H NMR data of all synthesized compounds. The coupling constants are not linked and the peak should be assigned (although, it is not necessary to specify in the paper). For example in the case of compound 1; From the result of the coupling constant, it can be said that the signals at 8.41 and at 8.27 are coupled (both J = 8.6), but they are 1H and 2H, respectively. These results are inconsistent.
- Page 18, Figure 10; the photo of bottom right (Merge) is wrong.
Author Response
- Page 4 to 5, 2,2,1 synthesis of starting material; The authors should recheck the 1H NMR data of all synthesized compounds. The coupling constants are not linked and the peak should be assigned (although, it is not necessary to specify in the paper). For example in the case of compound 1; From the result of the coupling constant, it can be said that the signals at 8.41 and at 8.27 are coupled (both J = 8.6), but they are 1H and 2H, respectively. These results are inconsistent.
Answer: Dear reviewer, thank you for your vigilance. We have corrected all spectra analysis.
- Page 18, Figure 10; the photo of bottom right (Merge) is wrong.
A: Figures has been corrected accordingly.
Reviewer 2 Report
In this manuscript Musioł et. al. report the synthesis and characterization of new mega Stokes quinoline based fluorophores with possible biological applications. The spectroscopic measurements are also backed by computational studies. Overall the manuscript is well presented, but some issues still need improvement, which are listed in the following:
- The introduction is focusing mainly on quinoline derivatives, but I think more large Stokes dyes should be cited. As the main goal seems to be the biological application for fluorescence microscopy, I recommend mentioning the importance of large Stokes dyes in that context, e.g. for multi-color imaging avoiding the problem of chromatic aberration (escpecially important in super resolution techniques, there is a review with a range of commercially available large Stokes dyes applicable in STED: 10.1088/2050-6120/3/4/042004, I strongly recommend adding at least this one as a new reference), or using as FRET donors minimizing cross talk.
- I find it confusing to restart compound numbering on figure 2.
- The syntheses are well described. I have some concern with 1H NMR interpretation, there are some overlapping peaks listed as dd, but from the unusual coupling constants I suspect those are two unrelated peaks having really close chemical shifts. If the peaks are overlapping, these can be listed as for example 8.02-7,99 (m, 2H, contained in this multiplet: 8.01 (d, J = 8 Hz, 1H) and 8.00 (d, J = 8 Hz, 1H)) - or simply as multiplet; or if they are just really close, but not overlapping, simply report them as separate peaks. These are my guesses without seeing the actual spectra:
Compound 1: 8.02: I guess this contains 2 signals each with J around 8 Hz, one 1H, other 2H
Compound 2: 7.91: two doublets with J around 8Hz; 7.69: without seeing the spectrum I cannot really interpret this signal, it might be a trans double bond hydrogen's doublet (J 16-18 Hz) overlapping with maybe a triplet from the quinoline (J 7-9 Hz)
Compound 3a: 7.87: two distinct doublets each with J 8.8Hz
Compound 3b: 7.57: this is a trans double bond doublet (J around 16-18 Hz) overlapping with an ortho doublet (J around 8Hz), the two outer peaks go for the trans double bond, the two inner peaks for the aromatic hydrogen
Compound 3c: 7.89 and 7.86: these are a trans double bond doublet (J around 16-18 Hz) overlapping with an ortho doublet (J around 8Hz), just they got mixed during the interpretation, the first and third peaks from left go for the aromatic hydrogen, and the second and third peaks for the trans double bond, I think
Compound 3d: 7.86: I cannot interpret this not seeing the spectrum, these might be two trans double bond hydrogens, or something even overlapping with the neighboring multiplet
Compound 3e: 7.63: this might be a trans double bond doublet (J around 16-18 Hz) overlapping with an ortho doublet (J around 8Hz), the two outer peaks go for the trans double bond, the two inner peaks for the aromatic hydrogen
Compound 3f: 7.89 singlet with a reported coupling constant? I think 7.89 and 7.87 may be overlapping signals, both have unusal coupling constants. - Table 1 could be augmented with molar extinction coefficients at the excitation wavelenths. There are fluorescent dyes with not too high quantum yields but high absorption giving a high brightness. So reporting the extinction coefficients also helps seeing the usefulness of the dyes.
- The compounds have large Stokes shifts, but also wide spectra (as usual for large Stokes shift dyes), meaning that there is some overlap of the absorption and emission spectra. Isn't the concentration used for fluorescence measurements (especially for quantum yield determinaton) too high?
Some aesthetic issues:
- The drawing style of the chemical structure changes even in the same figure (both figure 1 and 2)
- The colors on the spectra are somewhat hard to distinguish
Author Response
In this manuscript Musioł et. al. report the synthesis and characterization of new mega Stokes quinoline based fluorophores with possible biological applications. The spectroscopic measurements are also backed by computational studies. Overall the manuscript is well presented, but some issues still need improvement, which are listed in the following:
- The introduction is focusing mainly on quinoline derivatives, but I think more large Stokes dyes should be cited. As the main goal seems to be the biological application for fluorescence microscopy, I recommend mentioning the importance of large Stokes dyes in that context, e.g. for multi-color imaging avoiding the problem of chromatic aberration (escpecially important in super resolution techniques, there is a review with a range of commercially available large Stokes dyes applicable in STED: 10.1088/2050-6120/3/4/042004, I strongly recommend adding at least this one as a new reference), or using as FRET donors minimizing cross talk.
Answer: The introduction has been expanded by discussion on application of large stokes shift dyes in STED and FRET microscopy. Additional citations have been included as suggested.
- I find it confusing to restart compound numbering on figure 2.
A: numbering of the compounds has been changed.
- The syntheses are well described. I have some concern with 1H NMR interpretation, there are some overlapping peaks listed as dd, but from the unusual coupling constants I suspect those are two unrelated peaks having really close chemical shifts. If the peaks are overlapping, these can be listed as for example 8.02-7,99 (m, 2H, contained in this multiplet: 8.01 (d, J = 8 Hz, 1H) and 8.00 (d, J = 8 Hz, 1H)) - or simply as multiplet; or if they are just really close, but not overlapping, simply report them as separate peaks. These are my guesses without seeing the actual spectra:
Compound 1: 8.02: I guess this contains 2 signals each with J around 8 Hz, one 1H, other 2H
Compound 2: 7.91: two doublets with J around 8Hz; 7.69: without seeing the spectrum I cannot really interpret this signal, it might be a trans double bond hydrogen's doublet (J 16-18 Hz) overlapping with maybe a triplet from the quinoline (J 7-9 Hz)
Compound 3a: 7.87: two distinct doublets each with J 8.8HzCompound 3b: 7.57: this is a trans double bond doublet (J around 16-18 Hz) overlapping with an ortho doublet (J around 8Hz), the two outer peaks go for the trans double bond, the two inner peaks for the aromatic hydrogenCompound 3c: 7.89 and 7.86: these are a trans double bond doublet (J around 16-18 Hz) overlapping with an ortho doublet (J around 8Hz), just they got mixed during the interpretation, the first and third peaks from left go for the aromatic hydrogen, and the second and third peaks for the trans double bond, I thinkCompound 3d: 7.86: I cannot interpret this not seeing the spectrum, these might be two trans double bond hydrogens, or something even overlapping with the neighboring multipletCompound 3e: 7.63: this might be a trans double bond doublet (J around 16-18 Hz) overlapping with an ortho doublet (J around 8Hz), the two outer peaks go for the trans double bond, the two inner peaks for the aromatic hydrogenCompound 3f: 7.89 singlet with a reported coupling constant? I think 7.89 and 7.87 may be overlapping signals, both have unusal coupling constants.
A: Dear reviewer, thank you very much for you vigilance we have corrected all spectra analysis.
- Table 1 could be augmented with molar extinction coefficients at the excitation wavelenths. There are fluorescent dyes with not too high quantum yields but high absorption giving a high brightness. So reporting the extinction coefficients also helps seeing the usefulness of the dyes.
A: According to the Reviewer suggestion we added the molar extinction coefficients to Table 1 and the changes were done in the manuscript.
- The compounds have large Stokes shifts, but also wide spectra (as usual for large Stokes shift dyes), meaning that there is some overlap of the absorption and emission spectra. Isn't the concentration used for fluorescence measurements (especially for quantum yield determinaton) too high?
A: Thank You for your remarks, the overlap of the absorption and emission spectra in the case of our compounds and similar small molecules (for instance quinoline, terpyridine, carbazole derivatives) is rather typical phenomenon. Small differences between absorption edge and emission maxima can also be caused by the electronic nature of the substituents in molecules, especially in the case of donor-acceptor (D-A) architecture [Y. Liu et al., J. Lumin. 2015, 157, 249–256.; F.-S. Han et al., Tetrahedron 2008, 64, 9108–9116.; B. Czaplinska et al., Dyes Pigments 2017, 144, 119–132.; D. Zych et al., Chemistry Select 2017, 2, 8221–8233.; A. Slodek et al., Chemistry - A European Journal 2018, 24, 9622–9631.; C. Fan et al. Optical Materials 2017, 64, 489–495.]. The absorption and emission spectra as well as fluorescence quantum yield were measured in a solution with a concentration equal 1.25·10-5 M. A solution of compounds diluted to 10-5 M is commonly used to measure the optical properties of compounds, pointing out that in our case the concentration probably cannot affect the spectral overlap. We did tests with lower concentrations which did not affected overlapping but resulted in lower intensities.
Some aesthetic issues:
- The drawing style of the chemical structure changes even in the same figure (both figure 1 and 2)
A: we have corrected figures 1 and 2
- The colors on the spectra are somewhat hard to distinguish
A: All spectra figures have been corrected. The colors now are more vibrant and contrast.